# Systemic and Pulmonary Inflammation/Oxidative Damage: Implications of General and Respiratory Muscle Training in Chronic Spinal-Cord-Injured Patients

**DOI:** 10.3390/biology12060828

**Published:** 2023-06-07

**Authors:** Oscar F. Araneda, Cristián Rosales-Antequera, Felipe Contreras-Briceño, Marcelo Tuesta, Rafael Rossi-Serrano, José Magalhães, Ginés Viscor

**Affiliations:** 1Integrative Laboratory of Biomechanics and Physiology of Effort (LIBFE), Kinesiology School, Faculty of Medicine, Universidad de los Andes, Monseñor Álvaro del Portillo, Las Condes, Santiago 12455, Chile; 2Physical Medicine and Rehabilitation Unit, Clínica Universidad de los Andes, Santiago 8320000, Chile; carosales@miuandes.cl (C.R.-A.); rrossi@uandes.cl (R.R.-S.); 3Physiology Section, Department of Cell Biology, Physiology, and Immunology, Faculty of Biology, Universitat de Barcelona, 08028 Barcelona, Spain; gviscor@ub.edu; 4Laboratory of Exercise Physiology, Department of Health Science, Faculty of Medicine, Pontificia Universidad Católica de Chile, Av. Vicuña Mackenna #4860, Santiago 7820436, Chile; fcontrerasb@uc.cl; 5Millennium Institute for Intelligent Healthcare Engineering, Av. Vicuña Mackenna #4860, Santiago 7820436, Chile; 6Exercise and Rehabilitation Sciences Institute, School of Physical Therapy, Faculty of Rehabilitation Sciences, Universidad Andres Bello, Santiago 7591538, Chile; marcelo.tuesta@unab.cl; 7Laboratory of Metabolism and Exercise (LaMetEx), Research Centre in Physical Activity, Health and Leisure (CIAFEL), Laboratory for Integrative and Translational Research in Population Health (ITR), Faculty of Sport, University of Porto, 4200-450 Porto, Portugal; jmaga@fade.up.pt

**Keywords:** spinal cord injury, lung, inflammation, oxidative stress, respiratory muscle training

## Abstract

**Simple Summary:**

Damage to the spinal cord affects the voluntary control of skeletal muscles and also the control of the autonomic nervous system, thus affecting the cardiorespiratory system and the magnitude of the impact that is related to the level and complete or incomplete nature of the spinal cord injury. Associated with spinal cord damage are a sedentary lifestyle, increased adipose tissue, and low-grade inflammation/oxidative damage at the systemic level and in the lung. Lung malfunction has been recurrently described in this pathological context in both epidemiological and experimental studies. Physical exercise (PE) seems to be a potential therapeutic strategy, which involves a large mass of muscle tissue, thus potentially improving both lung functionality and the control of systemic and pulmonary inflammation. Furthermore, it is hypothesized that specific PE for the respiratory musculature per se or in association with other general exercise protocols may positively contribute to the function of this tissue.

**Abstract:**

Chronic spinal cord injury affects several respiratory-function-related parameters, such as a decrease in respiratory volumes associated with weakness and a tendency to fibrosis of the perithoracic muscles, a predominance of vagal over sympathetic action inducing airway obstructions, and a difficulty in mobilizing secretions. Altogether, these changes result in both restrictive and obstructive patterns. Moreover, low pulmonary ventilation and reduced cardiovascular system functionality (low venous return and right stroke volume) will hinder adequate alveolar recruitment and low O_2_ diffusion, leading to a drop in peak physical performance. In addition to the functional effects described above, systemic and localized effects on this organ chronically increase oxidative damage and tissue inflammation. This narrative review describes both the deleterious effects of chronic spinal cord injury on the functional effects of the respiratory system as well as the role of oxidative damage/inflammation in this clinical context. In addition, the evidence for the effect of general and respiratory muscular training on the skeletal muscle as a possible preventive and treatment strategy for both functional effects and underlying tissue mechanisms is summarized.

## 1. Introduction

The improvement in out-of-hospital and in-hospital management of acute and chronic spinal cord injury in recent decades has led to an increase in the number of years of survival [1,2]. This has refocused the concern on preventing and treating complications involving medical care and hospitalization to improve the quality of life of these patients [3]. The impact of this pathology is broad, ranging from psychological [4,5] to relating to the social environment [6], chronic pain [7], and sexual/reproductive disturbances [8]. This can lead to damage and malfunctioning in various organs secondary to the establishment of pathological phenomena such as obesity, diabetes, dyslipidemia, and cardiovascular disorders [9], on which, in recent years, research has focused, to the detriment of other medical problems [10,11,12,13]. Thus, the respiratory system has been little studied, even though many patients have clinical manifestations of this system and evidence of complications that affect the quality of life of the group in general, as well as, in the case of athletes, affect their physical performance [14,15]. Thus, in the following narrative review, we aim to describe the current evidence regarding both pulmonary effects, the role of localized inflammation/oxidative damage as a central mechanism of systemic complications, and the impact of pulmonary malfunction on physical performance. Finally, we summarize the literature that has studied the beneficial role of both general physical training involving the whole organism (particularly the locomotor musculature) and, in particular, the respiratory musculature for both athletic and non-athletic subjects with this pathology.

## 2. General Context of Spinal Cord Injury

Spinal cord injury consists of damage to the structural integrity of the spinal cord. When this happens, voluntary motor functions are affected, sensory information is lost, and it affects the automatic control of some organs provided by the autonomic nervous system [16]. From an epidemiological point of view, it is a condition that implies a high cost for the health system due to its prevalence. According to the WHO, between 250,000 and 500,000 people suffer a spinal cord injury annually, mainly associated with traffic accidents, falls, or acts of violence [17,18]. This condition’s organic repercussions are closely related to the spinal cord damage level. Thus, the closer to the brain it occurs, the greater the compromise of mobility and the effects on autonomic control, which implies a lower quality of life and an increase in complications, including compromised pulmonary function, cardiac conditions, vascular damage, loss of control of urination and defecation [19], as well as recurrent infectious complications (genitourinary, pressure ulcers, respiratory infections) and the presence of changes, both systemic and localized, in which inflammation and oxidative damage have been identified as participants in the origin and progression of tissue damage. 

## 3. Pulmonary Effects of Spinal Cord Injury

Spinal cord injury affects most of our body systems; however, a more detailed description of the pulmonary effects at rest and during exercise is of interest for this review. Thus, the respiratory muscles surround the thoracic cage and, in conjunction with the skeletal structure, establish changes in the magnitude and direction of intrapulmonary forces contributing to tissue deformation, favoring the entry and exit of air. To perform their function, the striated nature of the respiratory muscles requires the participation of motor innervation that may be affected in spinal cord injury. Changes in position as well as changes in the body and thoracic stability are added to the above. Thus, both postural control problems and loss of muscle strength specific to this segment will vary depending on injury level, restricting adequate air mobilization [15,16]. This phenomenon translates clinically into both decreased muscle strength (measured as peak inspiratory/expiratory pressure and respective flows) and decreased vital capacity and FEV1, which has been reported in multiple studies and linked to the decreased recruitment of alveolar units and the presence of atelectasis in these patients [20]. In addition, functional restrictions on defensive mechanisms such as cough and mucociliary clearance have been reported, which have been associated with an increased risk of lower respiratory infections [21]. Over time, due to their reduced participation in the ventilatory activity, atrophy of the respiratory muscles, a tendency to fibrosis, and decreased elasticity of the rib cage have been observed, resulting in an abnormal respiratory pattern that increases the ventilatory energy cost [22,23]. Another determinant of the magnitude of mobilized airflow corresponds to the resistance offered by the airway. In this aspect, spinal-cord-injured patients may show similar functional responses as patients classified as obstructive [24,25]. Several mechanisms explain this behavior; it has been shown that these patients’ airways have a decreased radius, partly due to lower respiratory muscle strength and lower thoracic excursion, as well as the difficulty in mobilizing secretions that affect the airway radius and, ultimately, in many cases, a compromise of the sympathetic nerves that correspond to the main bronchodilator factor through the adrenergic system of the airway’s smooth muscle [25]. In addition, the increased parasympathetic tone described in some patients further favors bronchial reactivity by increasing the work of breathing [26,27]. Ultimately, the poor pulmonary function described in this group decreases access to the large physiological pulmonary reserve, leaving patients unable to adequately respond to stressful situations. For example, this could be evidenced by greater needs such as physical exercise, exposure to hypoxia both environmentally and induced by clinical situations (alveolar infection, pulmonary embolism, surgeries, mechanical ventilation), as well as being more susceptible to contact with environmental pollutants or environments with low temperature or humidity that can trigger obstructive phenomena. A schematic summary of the changes described can be found in Figure 1.

## 4. Cardiorespiratory Response of Spinal Cord Injury Patients to Acute Exercise

Spinal cord injury negatively influences physical performance. This is evidenced by a decrease in the maximal capacity to perform aerobic physical exercise [28]. The mechanisms explaining this result are partly related to the decreased muscle mass (upper limb) involved in voluntary exercise. In these patients, a drop of 60 to 80% in VO_2_ max or peak has been described if exercise is performed with the arms relative to the legs [29]. From a cardiovascular point of view, a blood flow restriction to the tissues is influenced by a drop in heart rate [30], as well as in ejection volume, both due to a drop in venous return and also to a reduction in myocardial contractility [31], mainly due to a deficit in the functioning of the sympathetic nerves that emerge from the spinal cord [32,33]. Likewise, we find a lower blood volume [34], a drop in peripheral vascular resistance, and a deficit in the muscular activity of the lower limb that will limit venous return, which is the primary determinant of ventricular filling or preload and ejection volume [35]. As the sympathetic drive is affected, there will also be inefficient temperature control due to low sweating and an inadequate mobilization of metabolic fuels for exercise. To ameliorate this decrease in sympathetic action at the cardiovascular level, the induction of a state of autonomic dysreflexia via maneuvers known as “boosting” [36,37] or stimulation via electrical spinal cord stimulation [38] has been tried, while the use of functional electrical stimulation (FES) can promote venous return by contracting denervated muscles. In addition, oxygen extraction in the periphery will also be diminished due to the exclusive use of the arms and the usually severe muscle atrophy of the lower limb, which will imply a reduced capillarization and mitochondrial volume in these subjects [39,40].

Regarding the pulmonary factors that explain the decrease in maximal oxygen consumption, mechanical variables such as the position of the subject, less trunk control that may result in inefficient positions for a respiratory excursion, and a stiffer rib cage [41,42] may lead to relatively inefficient ventilatory behavior characterized by increased end-expiratory lung volumes (dynamic hyperinflation) limiting the increase in ventilation mainly at the expense of respiratory rate to the detriment of tidal volume, which may favor early respiratory muscle fatigue [43]. In addition, as mentioned in the previous paragraph, some patients also exhibit ventilatory characteristics analogous to an obstructive pattern that will contribute to an inability to adequately increase tidal volume due to a reduced ability to relax the airway smooth muscle [44]. It is also likely that the ventilatory component limits physical capacity by a lower pulmonary diffusion of oxygen, given that this organ will receive a lower relative blood flow during exercise and will also present more difficulty for the recruitment of alveolar units sited in the middle and upper lung zones, as the tidal volume is limited. Nevertheless, to our knowledge, this phenomenon has not been reported and can only be raised as a hypothesis to be studied. Finally, the changes observed in these patients are influenced by both the level (cervical, thoracic, lumbar) and the extent of spinal cord damage (complete or incomplete). Thus, regarding the cardiovascular and respiratory responses between quadriplegics and paraplegics, although they share some trends, such as lower values of venous return and ventricular function during exercise, it is evident that the magnitude of these changes has a greater magnitude in patients with cervical lesions [45]. To summarize the cardiorespiratory response to exercise in this group of patients, see Figure 2.

## 5. Systemic Inflammation in Spinal Cord Injury Patients

In most patients with spinal cord injury, there is concomitant damage to the blood–spinal-cord barrier, forming cellular debris and releasing intracellular proteins, constituting a potent pro-inflammatory stimulus [46]. Thus, during the first hours, this phenomenon activates natural immunity expressed by the action of microglia, in addition to neutrophilic infiltration that is associated with the release of pro-inflammatory mediators (IL-1, IL-6, IL-8, IL-12, TNF-α IFN-γ, CXCL1/CXCL12), the establishment of oxidative damage, and the activation of proteases that initiate the secondary damage associated with neural injury [47,48]. Subsequently, the zone will be infiltrated by tissue-damage-generating macrophages of the M1 type and then of the M2 type involved in reparative processes. In addition, the zone will be infiltrated by B and T lymphocytes, which, despite entering in low numbers, have a central role in regulating the inflammatory process [49,50]. The neuroinflammation described is relevant since it determines the magnitude of secondary damage and because an inflammatory response with aspects of chronicity may be established [51,52].

At the same time that nervous tissue inflammation occurs, a systemic inflammatory process begins to establish itself. The acute stage can be of great magnitude, to diminish and remain to a low degree beyond one year, in the period known as the chronic phase of the spinal cord injury [53]. In this phase, due to the lack of mobility, poor anabolic capacity is established in addition to mitochondrial malfunction, altered lipid profile, hyperglycemia, the appearance of fatty liver, and the change in body composition that favors the deposition of visceral fatty tissue. This last factor is relevant because fat tissue can release substances such as IL-6, plasminogen activator inhibitor-1, and TNF-α [54]. Systemic inflammation is considered an essential phenomenon because of the links that have been established with cardiac and vascular damage, alterations in immunity, and pulmonary dysfunction that constitute a significant source of complications and mortality in these patients [55].

The causes of this inflammatory phenomenon are still under debate; however, it has been established that it may have an infectious origin in pressure ulcers and genito-urinary tract infections and from the previously mentioned increase in fatty tissue post spinal cord injury. Regarding studies describing systemic inflammation, Segal et al., 1997, found higher plasma concentrations of IL-2R, ICAM-1, and IL-6 in spinal cord injury compared to healthy controls, and higher concentrations of these mediators when pressure ulcers were present in the spinal cord injury group [56]. Frost et al., 2005, also found increased C-reactive protein levels, with no changes in plasma IL-6 and TNF-α. In addition, he reported that the main factor associated with this phenomenon was the permanent use of urinary catheters [57]. Likewise, Morse et al., 2008, found that the plasma C-reactive protein concentration in these patients was associated with a higher presence of urinary tract infections and pressure ulcers. They also found positive associations between body mass index and the degree of movement difficulty [58]. Manns et al., 2005, found elevated plasma levels of IL-6 and CRP in paraplegic patients, and also significant correlations between plasmatic CRP versus total body fat and abdominal diameter [59]. Increased plasma concentrations of other markers have also been reported. Increases in endothelin-1 and sVCAM-1 [60] as well as increased FABP4 in the skeletal muscle and peripheral blood have been found in spinal cord injury patients compared to healthy controls [61]. From a therapeutic point of view, the infectious origin of this systemic inflammation can be specifically treated, but it is not completely suppressed, due to the metabolic component of the phenomenon. Consistently, although the effects of ibuprofen seem to lower the concentrations of C-reactive protein and plasma interleukin-6 in spinal cord injury patients [62], the coexistence of vascular damage pre-exclude the chronic use of ibuprofen for these purposes. Another therapeutic approach consists of dietary management [63] and physical exercise in its different modalities, which has shown favorable results in improving metabolic dysfunction associated with systemic inflammation [64,65,66].

## 6. Inflammation/Pulmonary Oxidative Damage in Spinal Cord Injuries

Just as systemic inflammation exists in spinal cord injury, it also gives rise to inflammation and localized oxidative damage in organs [55,67]. In the case of the lung, these phenomena play a central role as pathogenic factors in diseases such as asthma, inflammation in the infectious context, atelectasis, pulmonary hypertension, and cystic fibrosis [68,69]. In the case of spinal cord injury, few studies have explored this pathological phenomenon in this organ; Radulovic et al., 2010, compared nitric oxide in exhaled air in nine tetraplegics versus moderate asthmatics and healthy controls. He found that tetraplegic patients had values similar to asthmatics and higher than those of healthy controls [70], without evidence of spirometric changes (FEV1, FVC, and FEV1/FVC) in this group. In the other study conducted so far, a higher concentration of 8-isoprostanes, an indicator of lipoperoxidation, was found in tetraplegics compared to asthmatics and healthy controls in exhaled air condensate, with no change in this marker measured in plasma, suggesting that this phenomenon occurred at a localized level in the organ. In this case, tetraplegic patients had a restrictive spirometric pattern, with a lower FVC, FEV1, and ratio between these parameters in the normal range. Finally, the authors reported no association between isoprostane concentrations in the exhaled breath condensate and spirometric values [71].

## 7. Inflammation and Oxidative Damage in Acute and Chronic Exercise (Training)

A phenomenon commonly reported in conventional athletes when exercising is the establishment of an inflammatory process and oxidative imbalance, both in skeletal muscle, in the respiratory tract, and further at the systemic level. The magnitude of this phenomenon depends on the exercise intensity, duration, environmental conditions, and degree of training [72]. In the case of patients with chronic spinal cord injury, some studies have determined the appearance of a systemic inflammatory response/oxidative damage following acute exercise. Thus, in trained tetraplegics who performed 20 min of PE at 60% of VO_2_ max on an arm ergometer, a lower post-exercise plasma IL-6 release was found compared to normal controls, which was attributed to skeletal muscle atrophy [73]. In another report, Unemoto et al., 2011, found increases in plasma IL-6 concentration in thoracic spinal cord injury patients who performed a 2 h test at 60% of VO_2_ on an arm-crank ergometer, with no differences against healthy controls [74]. In patients with thoracic and lumbar spinal cord injury, Ogawa et al., 2014, found increases in plasma IL-6 in both groups after a wheelchair half marathon, with a significant increase in athletes with lumbar spinal cord injury. In addition, the cervical cord damage group reported a decrease in post-exercise plasma TNF-α [75]. In another report, Paulson et al., 2013, compared a submaximal test followed by a test to exhaustion in tetraplegic, paraplegic, and non-spinal-cord-injured wheelchair runners, with post-exercise IL-6 increases in the paraplegics and non-spinal-cord-injured group, but no change in the tetraplegics. No changes in plasma IL-10, IL-1ra, and TNF-α concentrations were found in either group [76]. Regarding inflammation/oxidative damage at the lung level, our research group found an increase when comparing before and after exercise of H_2_O_2_ (0.12 ± 0.09 vs. 0.23 ± 0.20 μM with a *p* = 0.036) and nitrite (1.53 ± 0.71 vs. 2.18 ± 1.33 μM with a *p* = 0.029) in exhaled breath condensate in quadriplegic wheelchair rugby players (ten males and two females) at the national team level after a total playing time of 22.0 ± 8.0 min at a perceived exertion of 4.0 ± 1.54 (unpublished data). In addition, no correlations were found between nitrite concentration and nitrogen concentration. Furthermore, no correlations were found between absolute changes in prooxidant concentrations and playing times or perceived exertion values nor with basal spirometric parameters; so far, there are no other reports of this phenomenon in this population.

Although paradoxical, the repetition of acute exercise, with the consequent increased production of pro-inflammatory and pro-oxidant factors during and after the acute event, can induce an anti-inflammatory and antioxidant state through a mechanism called hormesis [77]. Thus, physical exercise performed regularly in the context of physical training programs simultaneously generates anti-inflammatory mediators such as IL1-ra and IL-10, and TNF receptors, ultimately decreasing the acute inflammatory response to exercise. Furthermore, increased free radical generation in the context of systematic physical exercise will reduce the production of these species during acute exercise and, on the other hand, increase the antioxidant defenses through activation of the Keap1/Nrf2/ARE signaling pathway. Altogether, these mechanisms will contribute to cytoprotective effects by promoting the activation of detoxifying, anti-inflammatory, and antioxidant elements [78,79]. 

In this population, few reports have investigated the effect of systemic training on oxidative damage/inflammation. In this case, this question is of great importance because chronic low-grade inflammation is present in these patients, and exercise seems to be a therapeutic alternative for this deleterious condition. In this regard, Allison et al., 2016, reported benefits on physical and cardiovascular capacity without changes in plasma mediators, both pro- and anti-inflammatory (IL-1α, IL-1β, IL-4, IL-6, IL-8, IL-10, IL-13, CRP, MCP-1, TNF-α, and IFN-γ) by performing 12 weeks of training (three sessions per week) with FES cycling [80]. In another protocol, van Duijnhoven et al., 2010, conducted a training program in nine spinal-cord-injured subjects (about 40 years old) with cervical and thoracic injuries compared to nine healthy controls for eight weeks using an FES cycling device. The program lasted for eight weeks (twenty sessions), with the effect of the 30 min acute exercise on an arm bike being measured before and after the end of the training period. The researchers reported an inverse correlation between plasma MDA concentration, with no differences between baseline and post-acute-exercise values in plasma MDA concentration, superoxide dismutase, and intraerythrocytic glutathione peroxidase activity [81]. At the moment, there are no reports specifically measuring the effect of training on lung inflammation at the local level, but recently, Yates et al., 2022, described that after 12 weeks of training with FES of the rowing type, between 70 and 85% of the maximum heart rate frequency, for three sessions a week, found an improvement in lung function parameters, expressed as FEV1 and FVC and that these changes also correlated with changes in the plasma concentration of C-reactive proteins and leukocyte count [82]. Their finding is remarkable as it provides experimental evidence tying previous epidemiological associations between lung function (FEV1 and FVC) and systemic inflammation [83,84].

## 8. Respiratory Muscle Training for Spinal Cord Injury Patients: A Strategy to Improve Lung Function and Physical Performance: Can It Also Diminish Lung Inflammation?

Usually, the motor rehabilitation process begins early after spinal cord injury, progressively involving passive mobilization in the acute phase to active functional exercises against resistance. Although the main objective is locomotor restoration, it has been observed that its performance has an impact on the prevention and treatment of pulmonary complications [85]. Thus, Xiang et al., 2021, found improvements in motor activity and increases in FVC and FEV1 in spinal-cord-injured persons who participated in a rehabilitation program with an exoskeleton [86]. Because locomotor training impacts pulmonary function, it is currently recommended to complement this treatment with cardiopulmonary rehabilitation, using early specific respiratory exercises to preserve respiratory functionality [87], which is relevant since decreased pulmonary function in this population limits the tolerance to aerobic exercise, favoring the development of cardiometabolic disorders [85]. Pulmonary rehabilitation techniques range from assistance in secretion management to respiratory muscle training. Thus, several studies have demonstrated the effectiveness of this type of training on respiratory function, mainly when performed against a progressive load [88,89,90]. One study of note was conducted by Shin et al., 2019, in which 104 patients participated in a self-directed training program (incentive spirometer/glossopharyngeal breathing exercise/ air stacking exercise) lasting four to eight weeks in spinal cord injury patients of varying heights and time since injury, finding a partial recovery of lung volumes [91], which is relevant as these have been identified as predictors of respiratory infections [92].

Respiratory muscle activity generates enough alternating pressure differences with the atmosphere to promote air passage to and from the lung parenchyma, thus allowing gas exchange. These muscles are of the striated type and, therefore, are under control by alpha motor neurons, which will generate the appearance of muscle weakness when spinal cord damage is established in the upper cervical and thoracic segments. Due to this problem and the previously mentioned alterations to ventilatory mechanics, the possibility of improving the function of these muscles through training exists. In people without spinal cord injury, intense exercise alone activates these muscle groups; however, for both athletes and patients with lung diseases, specific training has shown improvements in their functionality expressed as increased strength, and improved lung volumes [93,94] and physical performance in athletes [95]. In this sense, it can be interesting to strengthen the respiratory musculature in patients with chronic spinal cord injury to prevent complications and favor tolerance or improve performance in Paralympic athletes (see Figure 1 and Figure 2). Thus, various training strategies have been designed. One method, almost exclusively tested in spinal-cord-injured patients, consists of electrical stimulation at the level of the thoracic and abdominal muscles, which, apart from favoring an increase in muscle strength to mobilize air, can also improve and affect the stability of the trunk, benefiting general ventilatory activity [96,97]. The other type of respiratory muscle training derives from the rehabilitation of patients with lung disease by increasing the work of breathing through increasing airway resistance, for which multiple devices have been developed by the researchers themselves and others are commercially available.

One way to increase resistance is by augmenting the dead space [98]. Regarding this training strategy, we did not find reports evaluating its use in patients with chronic spinal cord injury. However, at least theoretically, it may constitute a training method to explore in this group, as it offers the possibility of training at a low load of added resistance to the airway, which may favor treatment adherence and can also be used during exercise [99]. In contrast to these potential benefits, the main difficulty is to establish the optimal volume of dead space, the lack of commercially available equipment designed for this purpose, and the discomfort often experienced by users mainly due to possible hypercapnia [100]. As another strategy to increase respiratory muscle work, equipment was designed using a resistance valve system, which can train both the inspiratory and expiratory muscles, with its use being expanded to both athletes and patients without primary lung disease. In the case of spinal cord injury, in one of the first attempts to determine the response to respiratory muscle training in quadriplegics, Gross et al., 1980, reported increases in peak inspiratory pressure after eight weeks of training with a resistance valve [101]. In a similar population, Uji et al., 1999, found increases in peak oxygen consumption on an arm-crank ergometer with no differences in spirometry parameters and peak inspiratory pressure after training for six weeks [102]. Another report, in a group of paraplegics, compared respiratory muscle training versus conventional breathing exercises for four weeks, suggesting increases in their peak inspiratory and expiratory pressures and changes in functional tests indicative of improved physical performance [103]. There are few reports of respiratory muscle training in spinal-cord-injured athletes. Thus, West et al., 2013, compared a six-week inspiratory muscle training program versus the use of a bronchodilator in spinal-cord-injured rugby players. In the trained group, the author found increases in diaphragm thickness, peak inspiratory pressure, peak tidal volume and trends in deployed work, and VO_2_ peak on an arm-crank ergometer [104]. In another study, Lichte et al., 2008, found increases in peak inspiratory pressure after ten weeks of training with a resistance valve, with no effect on maximal voluntary ventilation and VO_2_ peak in athletes from different sports specialties with cervical to thoracic level injuries [105]. Finally, Gee et al. trained both inspiratory and expiratory musculature in wheelchair rugby competitors with cervical spinal cord injury for six weeks with a valve resistance equivalent between 60 and 80% of maximal inspiratory pressure and found increases in peak inspiratory and expiratory pressures and forced expiratory flow, with no change in resting lung volumes. In exercise, increases in tolerated load during the test and a trend toward an increase in VO_2_ max were found [106]. Another relevant point is that both methods of training the respiratory muscles (increased dead space or lowered radius) involve increased intrapulmonary pressure, so it is possible to suggest that their execution involves varying degrees of tissue mechanical stress, both for the airway and the lung tissue adjacent to it. This is relevant as an analogous situation, such as intense and prolonged exercise (more than 90 min) in healthy athletes, has previously shown increases in the generation of prooxidant substances (H_2_O_2_ and nitrites), while moderate exercise performed for 20 min by paraplegic athletes generated an increase in these species as described above. Taken together with the described data of higher basal levels of exhaled NO [70], this fact suggests that these patients are more susceptible to mechanical tissue stress, which should be further evaluated. Finally, and analogous to the decrease in inflammatory response/oxidative damage by general exercise, it should be tested whether training of the respiratory muscles by increasing resistance may be effective in decreasing this localized phenomenon in the airway, as has been seen in conventional long-distance athletes who are less responsive to acute exercise stimuli [107,108,109]. Due to the above, it is possible to hypothesize that the hormesis mechanism involved in the systematic exercise sessions may ultimately induce defenses that decrease the acute exercise response, which needs to be tested experimentally. For an overview of the process, see Figure 3.

## 9. Conclusions

Chronic spinal cord injury is a multifaceted medical problem that has yet to be thoroughly studied in terms of elucidating both the pathogenic mechanisms of complications and the effects of therapeutic strategies. Studies published to date show low-grade inflammation/oxidative damage that favors long-term damage to the cardiovascular system. However, the link between this pathogenic factor and the lung must be better characterized. This is partly due to the technical limitations in studying tissue phenomena at the pulmonary level, as invasive techniques (broncho-alveolar lavage, biopsies) would be required. For the time being, studies using other methods (exhaled air/EBC study) have shown that the lungs of these patients also show evidence of inflammation/oxidative damage that could extend from the upper airways to the alveoli, raising the question of whether this phenomenon reflects the systemic situation or is part of a localized process. On the other hand, research should also be expanded to improve our understanding of the effect that exercise can have on systemic/pulmonary inflammation. More research is needed, both on general physical exercise, involving as much skeletal muscle mass as possible (and activation of the rest of the organs), whether performed spontaneously or by using assisted muscle contraction. This can be provided by the use of FES in the limbs, as well as respiratory muscle training. Respiratory muscle training involves tissue stress to the airways and stimulates the pulmonary elastic component through the use of resistance valves or by applying devices that increase the dead space, which have shown benefits as physical performance enhancers in spinal-cord-injured athletes.

## Figures and Tables

**Figure 1 biology-12-00828-f001:**
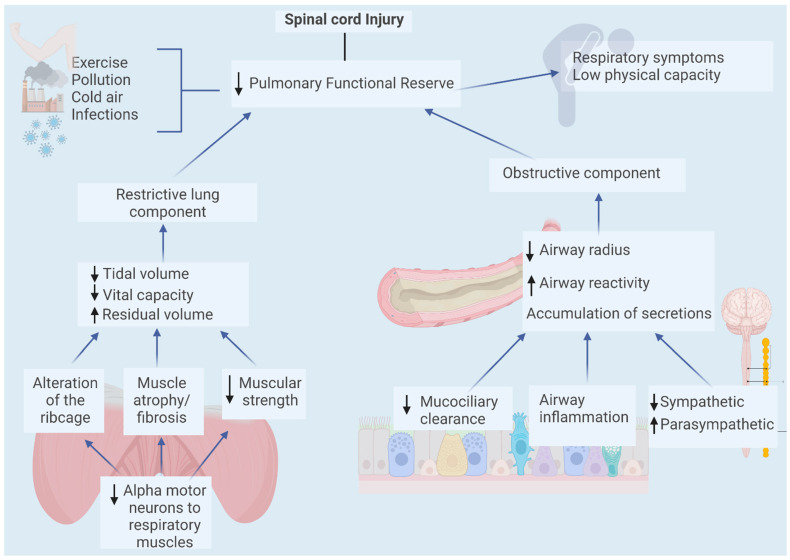
Pulmonary effects in spinal cord injury patients. The presence of pulmonary alterations is determined by both the level of the spinal cord injury and its involvement (complete or incomplete). Thus, patients may present obstructive or restrictive lung patterns or both, with a great variability in symptom presentation.

**Figure 2 biology-12-00828-f002:**
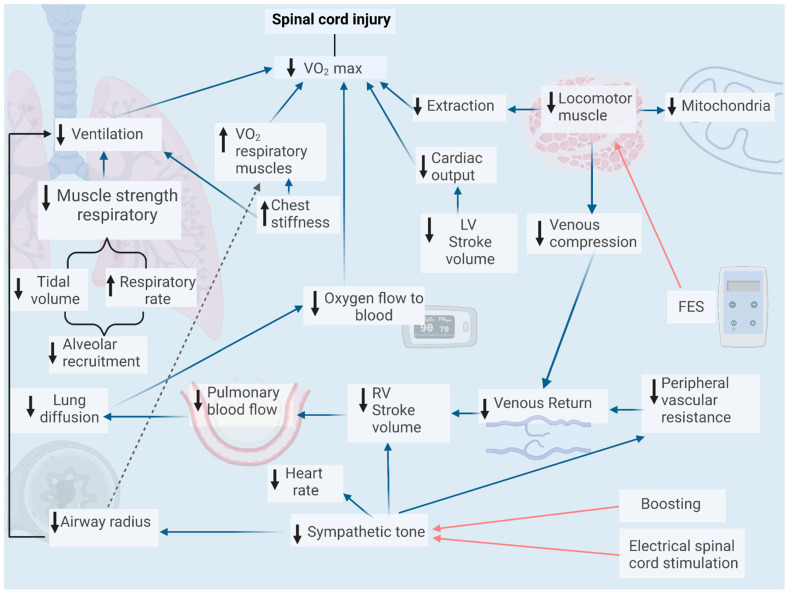
Summary of factors affecting cardiorespiratory malfunction during acute exercise and how this explains the decline in physical capacity in these patients.

**Figure 3 biology-12-00828-f003:**
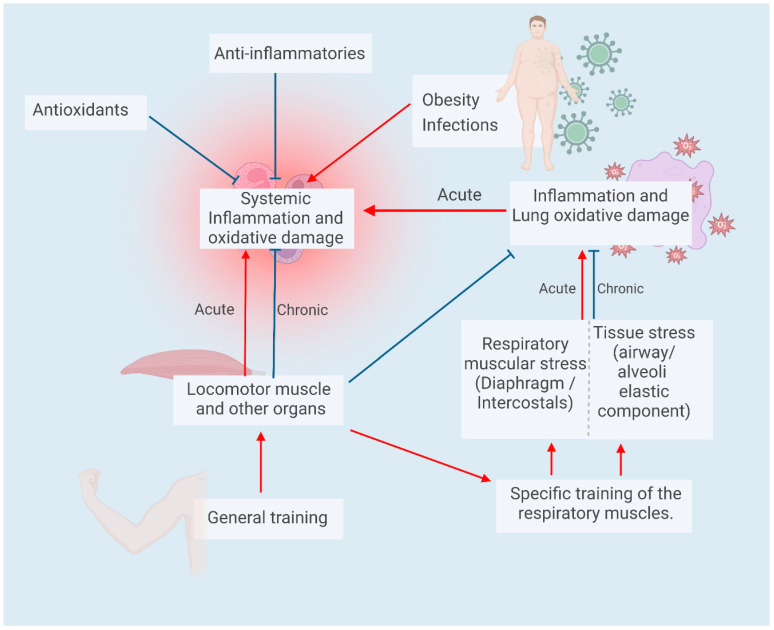
The proposed effect of general exercise (locomotor muscles and other organs) and specific muscle training on respiratory muscles decreases systemic and localized inflammation/oxidative damage.

## Data Availability

Not applicable.

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
