# Peer review of "Systemic and Pulmonary Inflammation/Oxidative Damage: Implications of General and Respiratory Muscle Training in Chronic Spinal-Cord-Injured Patients"

_biology, 2023, doi:10.3390/biology12060828_

Round 1
Reviewer 1 Report
The authors gave a nice review of the systemic and pulmonary inflammation post spinal cord injury (SCI). Authors also reported on the literatures on how acute and chronic exercise can alter the inflammatory/anti-inflammatory/oxidative molecules in SCI patients.
Specific comments:
In “Systemic inflammation in spinal cord injury patients” section, author might also want to comment about blood-brain barrier (BBB) disruption in SCI? The BBB disruption allows immune cells and inflammatory molecules to enter the central nervous system and perpetuate the inflammatory response.
Line 251, typo for TNF-α
Line 259-261, can author please show data to backup this statement?
Line 313 typo
References #1, 27, 97-99 are empty on the reference list
Author Response
The authors gave a nice review of the systemic and pulmonary inflammation post spinal cord injury (SCI). Authors also reported on the literatures on how acute and chronic exercise can alter the inflammatory/anti-inflammatory/oxidative molecules in SCI patients.
Specific comments:
In “Systemic inflammation in spinal cord injury patients” section, author might also want to comment about blood-brain barrier (BBB) disruption in SCI? The BBB disruption allows immune cells and inflammatory molecules to enter the central nervous system and perpetuate the inflammatory response.
A: Thank you for your comment. We have added a comment in this paragraph regarding the issue suggested by the reviewer.
Line 251, typo for TNF-α
A: Thank you for your comment. We have fixed this error.
Line 259-261, can author please show data to backup this statement?
A: At the reviewer's request we have added the data in the text.
Line 313 typo
A: The requested change was made
References #1, 27, 97-99 are empty on the reference list
A: Thank you very much for the comment. In the new version, this problem was fixed.
Reviewer 2 Report
The article entitled “Systemic and pulmonary inflammation/oxidative damage: Im-2 plications of general and respiratory muscle training in chronic 3 spinal cord-injured patients” describes both the deleterious effects of chronic spinal cord injury on the functional effects of the respiratory system as well as the role of oxidative damage/inflammation in this clinical context. The manuscript is well-written. The only question is where Fig 3 comes from. It seemed that the data were from the authors’ research group. If the data have been published, the authors should indicate it in the paper. Otherwise, I do not feel that it is proper to include figure containing unpublished data in a review.
Author Response
The article entitled “Systemic and pulmonary inflammation/oxidative damage: Im-2 plications of general and respiratory muscle training in chronic 3 spinal cord-injured patients” describes both the deleterious effects of chronic spinal cord injury on the functional effects of the respiratory system as well as the role of oxidative damage/inflammation in this clinical context. The manuscript is well-written. The only question is where Fig 3 comes from. It seemed that the data were from the authors’ research group. If the data have been published, the authors should indicate it in the paper. Otherwise, I do not feel that it is proper to include figure containing unpublished data in a review.
A: Thank you for your comment. At the reviewer's request we have extracted figure 3.
Reviewer 3 Report
In this narrative review, Araneda OF et al discuss pathophysiology of pulmonary function decline and related inflammation in patients with spinal cord injury, and the potential for exercise-based interventions to mitigate the risk of pulmonary pathology in this patient population. The authors have discussed the underpinning pathophysiologic mechanisms in great detail. Suggestions to improve manuscript:
1) In figures 1 & 2, please show spinal cord injury as the starting point in the mechanistic models since these pathways are being discussed in relation to spinal cord injury alone.
2) One important topic missing is the role of pulmonary / cardiopulmonary rehabilitation therapy. Personalized exercise regimens are a part of rehab therapy. Hence literature related to rehab should be discussed under its own separate sub-heading in the manuscript.
The manuscript is comprehensible.
Author Response
In this narrative review, Araneda OF et al discuss pathophysiology of pulmonary function decline and related inflammation in patients with spinal cord injury, and the potential for exercise-based interventions to mitigate the risk of pulmonary pathology in this patient population. The authors have discussed the underpinning pathophysiologic mechanisms in great detail. Suggestions to improve manuscript:
- In figures 1 & 2, please show spinal cord injury as the starting point in the mechanistic models since these pathways are being discussed in relation to spinal cord injury alone.
A: Thank you for your comment. As per the reviewer's request we have added the spinal cord injury in both figures.
- One important topic missing is the role of pulmonary / cardiopulmonary rehabilitation therapy. Personalized exercise regimens are a part of rehab therapy. Hence literature related to rehab should be discussed under its own separate sub-heading in the manuscript.
A: Thank you for your comment. We have added a paragraph regarding rehabilitation at the reviewer's request.